# Natural selection drives the evolution of mitogenomes in *Acrossocheilus*

**Dan Zhao**◉, **Yudong Guo**◉, **Yang Gao**◉*

Fisheries College, Zhejiang Ocean University, Zhoushan, Zhejiang, China

◉ These authors contributed equally to this work.
* gaoyang-82@163.com

**Data Availability Statement:** All relevant data are within the article and its Supporting information files.

**Funding:** The authors received no specific funding for this work.

## Abstract

The mitochondrial genome plays a crucial role in the balance of energy and heat production in organisms and, thus, may be under natural selection due to its potential role in adaptive divergence and speciation. Here, we studied natural selection on the mitogenome of *Acrossocheilus* (Cypriniformes: Cyprinidae), a genus of fish that inhabits a broad latitudinal distribution ranging from the tropics and subtropics through temperate regions. Specifically, we used 25 published mitogenome sequences of *Acrossocheilus* species to investigate phylogenetic relationships in this genus and detected signals of positive selection on 13 protein-coding, mitochondrial genes. We found that relaxed purifying selection and genetic drift were the predominant evolutionary forces acting on the analyzed mitogenomes. However, we also found evidence of diversifying selection on some codons, indicating episodes of positive selection. Additionally, we analyzed the mitogenomic data within an environmental modeling framework and found that the Ka/Ks ratio of *ATP6* may correlated with a mean diurnal temperature range (p = 0.0449), while the Ka/Ks ratio of *COX2* may correlated with precipitation during the driest month (p = 0.00761). These results suggest that the mitogenomes of *Acrossocheilus* species may be involved in evolutionary adaptations to different habitats. Based on this, we believe that our study provides a new insight into the role of the mitochondrial genome of *Acrossocheilus* species in adaptation to different environments. During our study, we also discovered several cases of paraphyly and polyphyly among accessions of species and their putative synonyms. Thus, our study suggests that a careful reassessment of the taxonomy of *Acrossocheilus* is using high-quality molecular data merited.

## Introduction

Mitochondria, which encode 22 transfer RNAs (tRNAs), two ribosomal RNAs (rRNAs), and 13 structural proteins in vertebrates, supply about 95% of adenosine triphosphate (ATP) through oxidative phosphorylation (OXPHOS), or electron transport [1]. Of the 37 total genes in the vertebrate mitochondrial genome (mitogenome), 13 encode core structural subunits of four out of five protein complexes (complexes I, III, IV, and V) that are directly involved in electron transport and ATP synthesis, and these two processes may directly drive the evolution of

**Competing interests:** The authors have declared that no competing interests exist.

the mitogenome. In particular, mitochondrial OXPHOS generates both energy and heat, which must be balanced with ATP production. Therefore, heat from the surrounding environment may represent a constraint on OXPHOS capacity at the genetic level [2, 3]. Consequently, the mitogenome and, especially genes involved in electron transport, may have crucial adaptive roles in the speciation of organisms [4, 5].

Selective pressures likely act on mitochondrial energy metabolism to drive adaptation of organisms to the different energy requirements within their unique environmental niches [6]. In fact, metabolic rate is known to govern both genetic divergence and speciation, which both show the same exponential relationship to environmental temperature [7]. Consequently, perhaps unsurprisingly, speciation often occurs in response to realized ecological opportunity and the evolution of habitat preferences [8]. Nevertheless, given strong evolutionary constraints on maintaining and balancing the energy needs of organisms, purifying selection may be the predominant form acting at the mitogenomic level [9]. However, several studies have revealed candidate sites undergoing positive selection within the protein-coding genes of the mitogenomes of vertebrates, such as in mammals [5], birds [10], and fish [11]. Thus, selection may operate similarly across organisms in the tree of life, largely in a purifying manner, but also drives adaptations in different genes as organisms expand their ranges into new environments through evolutionary time.

The fish genus, *Acrossocheilus* Oshima, 1919, provides a highly suitable model system for studying the possible roles in adaptive divergence of natural selection acting on the mitogenome. This genus has 26 species, which have a center of diversity (21 species) in southern China and commonly occur in middle and/or lower reaches of river drainage. Our study included 14 species of the genus *Acrossocheilus*. The remaining species, although reported, are difficult to sample and accurately identify due to their rarity and taxonomic confusion [12]. Therefore, only 25 published mitochondrial genome sequences from 14 species in this genus were included in this study. At the genus-level, *Acrossocheilus* exhibits a broad latitudinal distribution from the tropics through subtropical and temperate regions, but species tend to be more narrowly distributed, occurring in only one of these environmental types (e.g., *A. jishouensis*) [13]. In china, five species (*A. iridescens*; *A. malacopterus*; *A. spinifer*; *A. xamensis*; *A. ikedai*) inhabit the tropics, fourteen species (*A. microstoma*; *A. longipinnis*; *A. jishouensis*; *A. multi-striatus*; *A. parallens*; *A. rendahli*; *A. wenchowensis*; *A. wuyiensis*; *A. yunnanensis*; *A. paradoxus*; *A. hemispinus*; *A. clivosius*; *A. beijiangensis*; *A. kreyenbergii*) are distributed in the subtropical zone and two species (*A. Monticola*; *A. fasciatus*) occur in temperate regions. As fish, the genus comprises ectothermic species, such that their body temperatures and metabolic rates are highly dependent on environmental temperatures and other extrinsic parameters [14]. Therefore, the evolutionary adaptation of *Acrossocheilus* species to tropical, subtropical, or temperate environments likely involves selection on their mitogenomes [5, 15].

Here we generated a phylogenetic framework of *Acrossocheilus* using mitogenomes and used it for detecting signals of selection on mitochondrial protein-coding genes. Additionally, we used generalized linear models to determine the relationship between signatures of selection and parameters of the environments occupied by species. We believe that our study provides new insights on the evolutionary and biogeographic history of these fish and also elucidates specific ways in which mitogenomic evolution may drive macro-evolution of these and other organisms.

## Methods and materials

### Sequences

Our dataset consisted of sequences of 25 previously published mitogenomes representing 14 *Acrossocheilus* species (Table 1). Among the 14 species, ten (20 sequences) occur in subtropical

**Table 1. Details of the sequences used in this study.**

| Species names | Accession number | Locality |
|---|---|---|
| *Acrossocheilus barbodon* | NC_022184.1 | 109.49˚E, 19.20˚N |
| *Acrossocheilus beijiangensis* 1 | NC_028206.1 | 113.48˚E, 24.70˚N |
| *Acrossocheilus beijiangensis* 2 | KY131976.1 | 108.89˚E, 25.77˚N |
| *Acrossocheilus fasciatus* | NC_023378.1 | n.a. |
| *Acrossocheilus hemispinus* | NC_022183.1 | 118.75˚E, 26.39˚N |
| *Acrossocheilus iridescens* | NC_031551.1 | n.a. |
| *Acrossocheilus jishouensis* | NC_034917.1 | 109.15˚E, 26.63˚N |
| *Acrossocheilus kreyenbergii* 1 | NC_024844.1 | n.a. |
| *Acrossocheilus kreyenbergii* 2 | KY094969.1 | 108.9˚E, 25.78˚N |
| *Acrossocheilus longipinnis* | NC_047455.1 | 108.9˚E, 25.75˚N |
| *Acrossocheilus monticola* 1 | KT367805.1 | n.a. |
| *Acrossocheilus monticola* 2 | NC_022145.1 | n.a. |
| *Acrossocheilus paradoxus* 1 | AP009303.1 | n.a. |
| *Acrossocheilus paradoxus* 2 | MG878098.1 | n.a. |
| *Acrossocheilus parallens* 1 | AP011251.1 | n.a. |
| *Acrossocheilus parallens* 2 | KT715479.1 | 113.77˚E, 23.75˚N |
| *Acrossocheilus parallens* 3 | NC_026973.1 | 116.70˚E, 25.53˚N |
| *Acrossocheilus parallens* 4 | KP257293.1 | n.a. |
| *Acrossocheilus spinifer* | NC_034918.1 | n.a. |
| *Acrossocheilus stenotaeniatus* | NC_024934.1 | n.a. |
| *Acrossocheilus wenchowensis* 1 | NC_020145.1 | 120.51˚E, 27.82˚N |
| *Acrossocheilus wenchowensis* 2 | KC495074.1 | 118.03˚E, 29.82˚N |
| *Acrossocheilus wuyiensis* | NC_034919.1 | 118.06˚E, 27.78˚N |
| *Acrossocheilus yunnanensis* 1 | NC_028527.1 | 103.01˚E, 29.99˚N |
| *Acrossocheilus yunnanensis* 2 | MN395748.1 | 105.42˚E, 28.35˚N |
| *Onychostoma meridionale* | NC_031603.1 | n.a. |
| *Onychostoma barbatulum* | AP009311.1 | n.a. |

regions, two species (two sequences) inhabit the tropics, and two (three sequences) are distributed in the temperate zone. We also obtained the complete mitogenome sequences of *Onychostoma meridionale* and *Onychostoma barbatulum*, representing the sister genus to *Acrossocheilus* [16], to constitute the outgroup in phylogenetic analyses. From the mitogenomes of each species, we obtained the 13 annotated protein-coding genes for phylogenetic analysis and analyses of evolutionary rates, and aligned them in in MEGA 6.06 [17].

## Rates of evolution

We used alignments of the 13 protein-coding genes in *Acrossocheilus* (excluding the outgroup) to determine the number of polymorphic sites (S), the number of haplotypes (Nh), and theta ($\theta$) according to the finite sites model [18] in DnaSP v5.1 [19]. We used the $\theta$-values to infer the relative per-generation mutation rates ($\mu_{Relative}$) of individual of genes ($\mu_{gene}$) relative to the whole mitogenome ($\mu_{genome}$) using the equation $\theta = 4Ne\mu$, where population size (Ne) can be assumed to be similar for each gene, because population size is a property of each sequence in this case, and recombination is almost nonexistent in the mitochondrial DNA. We also determined the number of nonsynonymous and synonymous substitutions using TreeSAAP [20].

To assess neutral evolution in *Acrossocheilus* species, we performed an F test for normality of gene length, nonsynonymous and synonymous mutations, and the total number of

mutations, and then conducted linear regression analyses in PAST [21] using alignments for all 13 protein-coding genes. The analyses consisted of (1) the number of mutations versus alignment length in bases, (2) the number of synonymous mutations versus alignment length in bases, (3) the number of nonsynonymous mutations versus the alignment length in bases, and (4) the number of nonsynonymous changes versus synonymous changes.

## Phylogenetic analysis

Based on the concatenated alignment of the 13 protein-coding genes with the outgroup taxa included, we inferred phylogenetic relationships using maximum-likelihood (ML) in RAxML v8.2.4 [22]. Prior to performing the ML analysis, we determined that GTR+G was the best-fit model of nucleotide substitution in JMODELTEST v2.1.1 [23] under the Akaike Information Criterion (AIC). In RAxML, we calculated branch support via 1000 ML bootstrap replicates. Two *Onychostoma* species were used as outgroup. We used the resulting phylogeny as a framework for subsequent codon-based tests for selection.

## Analysis of positive selection

To investigate whether positive or purifying selection had occurred at the protein level, we calculated the ratio of nonsynonymous (Ka) to synonymous (Ks) (Ka/Ks > 1, positive selection; Ka/Ks = 1, neutrality; Ka/Ks < 1 negative or purifying selection) using the alignment of the 13 protein-coding genes of *Acrossocheilus* species. Thereafter, we determined the average pairwise Ka/Ks between sequences for each gene in FEL (Fixed Effects Likelihood) implemented on the Datamonkey server (http://www.datamonkey.org/dataupload.php).

In addition, we used phylogeny-based methods to detect signatures of positive selection at the codon level [24]. This approach facilitates identifying possible positive selection on a small number of codons within genes that may otherwise be masked by strong purifying selection. Specifically, we applied our reconstructed ML phylogenetic tree in CodeML of the PAML4 package [25] to perform a ML analysis of positive selection on codons. For the analysis, we compared several pairs of null and alternative models (M7 vs. M8, M1a vs. M2a, and M8a vs. M8) using a likelihood ratio test (LRT) to determine whether there are variable ratios of ω at particular codon positions [25] (S1 Table). These codon substitution models, which can be compared using likelihood-ration test, assume that the ω ratio is the same across branches of the phylogeny but different among sites in a multiple sequence alignment. We can identify positively selected codons if M2a (positive selection) provides a better fit than M1a (nearly neutral), or if M8 (beta and ω > 1) provides a better fit than M7 (beta) or M8a (beta and ω = 1). The M7 vs. M8 comparison offers a very stringent test of positive selection [26], while the M8a vs. M8 comparison yields fewer false positives [27].

In addition to testing models in CODEML, we used HyPhy [28] implemented on the webserver Datamonkey [29] to infer codons under selection according to the following approaches: Fast Unbiased Bayesian AppRoximation (FUBA) [30], Mixed Effects Model of Evolution (MEME) [31], Single Likelihood Ancestral Counting (SLAC), and Fixed-Effects Likelihood (FEL) [32]. For each approach, we applied the best-fit substitution model for each gene and assessed significance according to posterior probability > 0.9 (FUBAR) or P-value < 0.05 (MEME). After applying all approaches, we looked among them for agreement on sites under positive selection in order to reduce false positives [32].

To supplement the codon-based approaches, we also used TreeSAAP [20], which estimates significant changes in amino acid properties along the phylogeny. When amino acid substitutions have strong effects on protein biochemistry, they are considered candidates for selection. We ran the analysis with a sliding window size of 15 codons and a step size of 1 codon. We

considered amino acid substitutions as possibly under selection if the magnitude of change was ≥ 6, and we regarded z-scores above 3.09 or below– 3.09 (P < 0.001) as attributable to positive and purifying selection, respectively [33]. We applied these relatively stringent criteria to avoid detection of false positives.

## Environmental analysis

We applied a generalized linear model (GLM) to investigate the relationship between pairwise Ka/Ks values and environmental distances among accessions. We obtained environmental data for 14 of the 25 mitogenome accessions of *Acrossocheilus* (Table 1) with available georeferencing for their collection localities. Specifically, we used DIVA-GIS 7.4.01 [34] to extract values for the 19 standard bioclimatic variables (S2 Table) of WORLDCLIM 1.3 [35] and converted them to a distance matrix for the accessions in R 4.0.3. Bioclimatic variables are derived from the monthly temperature and rainfall values in order to generate more biologically meaningful variables, and are often used in species distribution modeling and related ecological modeling techniques. We obtained the distance matrix of pairwise Ka/Ks in DnaSP v5.1 [19] and performed the GLM using the LME4 library [36] for R 4.0.3 [37].

## Results

### Rates of evolution

Relative per-generation mutation rates ($\mu_{Relative}$) varied across mitochondrial genes (Table 2). Overall, genes of the *ND* family evolved faster than the other ones, with *ND6* evolving the fastest ($\mu_{Relative}$ = 1.61, Table 2). The most slowly evolving gene was *COX3* ($\mu_{Relative}$ = 0.9, Table 2). Across all genes, the number of synonymous substitutions was 2.5–12.7 times higher than nonsynonymous substitutions, and the Ka/Ks ratios for each gene were all < 1.0, suggesting signatures of purifying selection (Table 2 and Fig 1). The highest average pairwise nonsynonymous substitutions were in *ATP8* and *ND6* where Ka/Ks = 0.123 (Table 2 and Fig 1), and the lowest was found in *COX1* with Ka/Ks = 0.00899 followed by *CYTB*, *COX3*, and *ND4* (with values between 0.0184 and 0.0261) (Table 2 and Fig 1). In general, the ND gene family showed relatively high Ka/Ks compared to other genes.

We determined that gene length, nonsynonymous and synonymous mutations, and the total number of mutations were all normally distributed variables based on F-tests. Among these variables, our linear regressions revealed that the total number of mutations and synonymous and nonsynonymous changes were significantly correlated with gene length (Fig 2A, 2B and 2D). Additionally, the number of synonymous mutations showed a significant linear relationship with the number of nonsynonymous mutations (Fig 2C).

### Phylogenetic analysis

The ML phylogeny shows strong support for all relationships (bootstrap value ≥ 95, Fig 3). *Acrossocheilus monticola* and *A. yunnanensis* formed a clade as did *A. longipinnis* and *A. iridescens*. The latter clade was sister to *A. barbodon*. Further, we found that *A. hemispinus* was nested within the clade of *A. parallens* that was sister to a clade containing *A. jishouensis* and one of two individuals of *A. paradoxus*. *A. wenchowensis* was paraphyletic and included *A. fasciatus*, and this clade was sister to *A. kreyenberfii*. Both accessions of *A. beijiangensis* clustered together and were sister to *A. stenotaeniatus* and *A. spinifer*. This whole clade of three species was sister to *A. wuyiensis* and the second *A. paradoxus* individual. Thus, *A. paradoxus* was polyphyletic.

**Table 2. Summary statistics for the *Acrossocheilus* mitogenome sequences and regions.**

| Gene | Length | Nh | S | θ | μ_Relative | Non-Synonymous | Synonymous | Ka/Ks |
|---|---|---|---|---|---|---|---|---|
| ATP6 | 681 | 23 | 203 | 0.09 | 1.09 | 70 | 343 | 0.0391 |
| ATP8 | 165 | 20 | 38 | 0.06 | 0.73 | 15 | 42 | 0.123 |
| CYTB | 1140 | 24 | 365 | 0.10 | 1.18 | 85 | 699 | 0.0184 |
| COX1 | 1548 | 25 | 437 | 0.09 | 1.05 | 73 | 930 | 0.00899 |
| COX2 | 690 | 22 | 204 | 0.09 | 1.02 | 59 | 275 | 0.0436 |
| COX3 | 783 | 24 | 203 | 0.08 | 0.90 | 39 | 345 | 0.0213 |
| ND1 | 972 | 24 | 329 | 0.11 | 1.23 | 96 | 628 | 0.0310 |
| ND2 | 1044 | 24 | 403 | 0.12 | 1.42 | 186 | 648 | 0.0469 |
| ND3 | 348 | 23 | 124 | 0.11 | 1.24 | 42 | 207 | 0.0512 |
| ND4 | 1380 | 24 | 455 | 0.10 | 1.16 | 142 | 806 | 0.0261 |
| ND4L | 294 | 23 | 90 | 0.09 | 1.07 | 20 | 153 | 0.0319 |
| ND5 | 1824 | 24 | 616 | 0.10 | 1.20 | 244 | 982 | 0.0483 |
| ND6 | 519 | 24 | 217 | 0.14 | 1.61 | 121 | 297 | 0.123 |
| Mitogenome | 14319 | 25 | 4030 | 0.09 | 1.00 | | | |

The Ka/Ks is the average values of all pairwise comparisons.

Nh, number of haplotypes; S, number of polymorphic sites; θ, mutation rate.

## Analysis of positive selection

Based on the analyses in CodeML, LRT found no significant difference for the models M7 vs. M8 (S1 Table), and comparisons between M1a and M2a as well as M8a and M8 revealed no significant evidence for positive selection in *Acrossocheilus* species. Thus, taken together, the CodeML analyses did not uncover any signal of positive selection.

In contrast to CodeML, we detected codons under positive selection according to the MEME, FUBAR, SLAC, and FEL algorithms. Specifically, we found six codons under positive

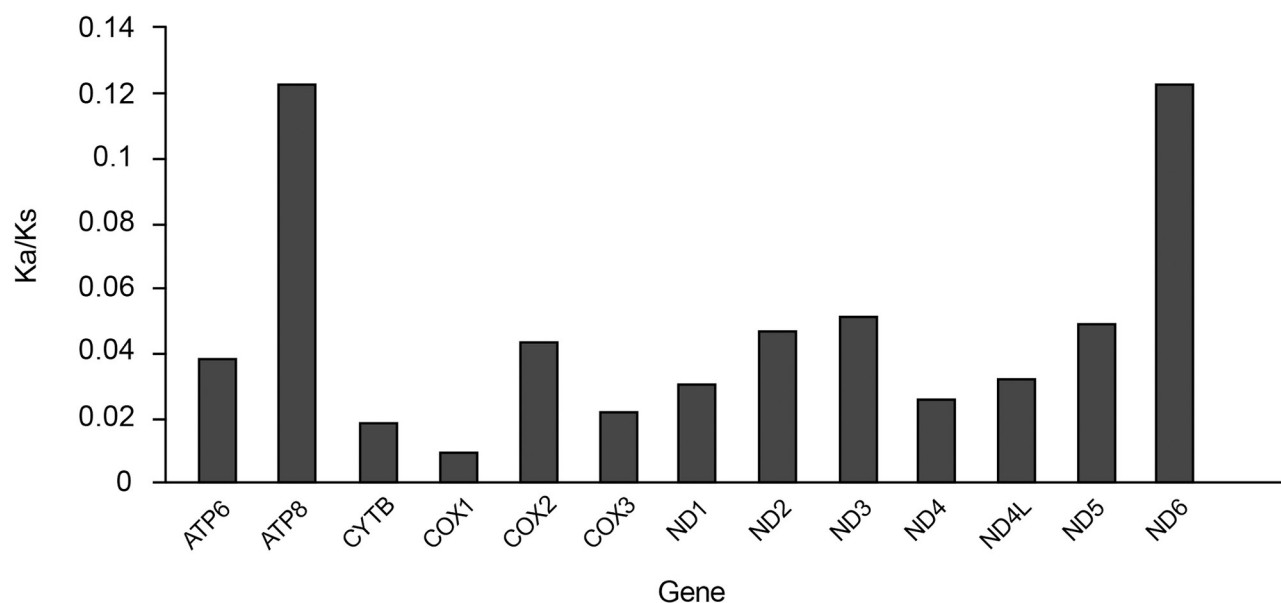

**Fig 1. The Ka/Ks ratios of the 13 different mitochondrial genes in *Acrossocheilus* species.**

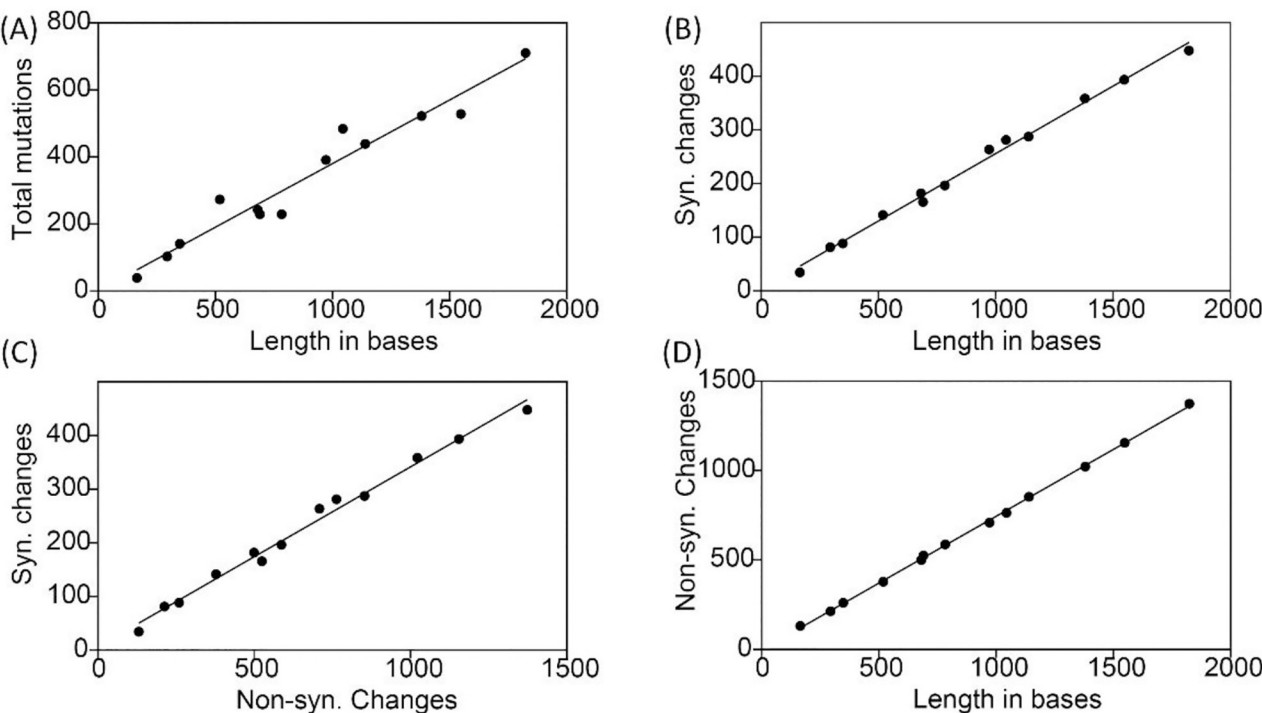

**Fig 2. Correlation analyses.** The dots denote the individual values for the 13 different mitochondrial genes in *Acrossocheilus* species and the line the best-fitted line. (A) Correlation between total number of mutations and length in bases of the genes. (B) Correlation between synonymous mutations and length in bases of the genes. (C) Correlation between nonsynonymous and synonymous mutations. (D) Correlation between nonsynonymous mutations and length in bases of the genes.

selection in five genes (*COX1*, *COX2*, *COX3*, *ND3*, and *ND4*) based on MEME, two codons under positive selection in two genes (*ND3* and *ND4*) using FUBAR and FEL, and only one codon in *ND4* under positive selection based on SLAC (Table 3). Codon 1 located within *ND3* was detected by all approaches except for SLAC, while the FUBAR, SLAC, and FEL algorithms agreed that 51 in *ND4* is involved in positive selection (Table 3). All other codons implicated in positive selection were detected by only one of the four approaches.

The results from TreeSAAP showed that 62 sites had changed considerably in amino acid properties (S3 Table). Across all these sites, 23 of them were under positive selection, while 39 showed signatures of purifying selection. One amino acid property, the equilibrium constant (i.e., ionization of COOH; S3 Table), was found, in general, to be under positive selection in all genes.

## Environmental analysis

Our GLMs revealed that the Ka/Ks ratios of two genes, *ATP6* and *COX2*, have significant relationships to the environment. In particular, there was a relationship ($p = 0.0449$) between the Ka/Ks ratio of *ATP6* and mean diurnal temperature range (Bio2; mean of monthly (max temp–min temp)). In *COX2*, the Ka/Ks ratio was correlated with precipitation during the driest month (Bio14), precipitation seasonality (Bio15; coefficient of variation), precipitation during the driest quarter (Bio17), and precipitation during the coldest quarter (Bio19). Among these, the highest significant correlation ($p = 0.00761$) was with precipitation during the driest month.

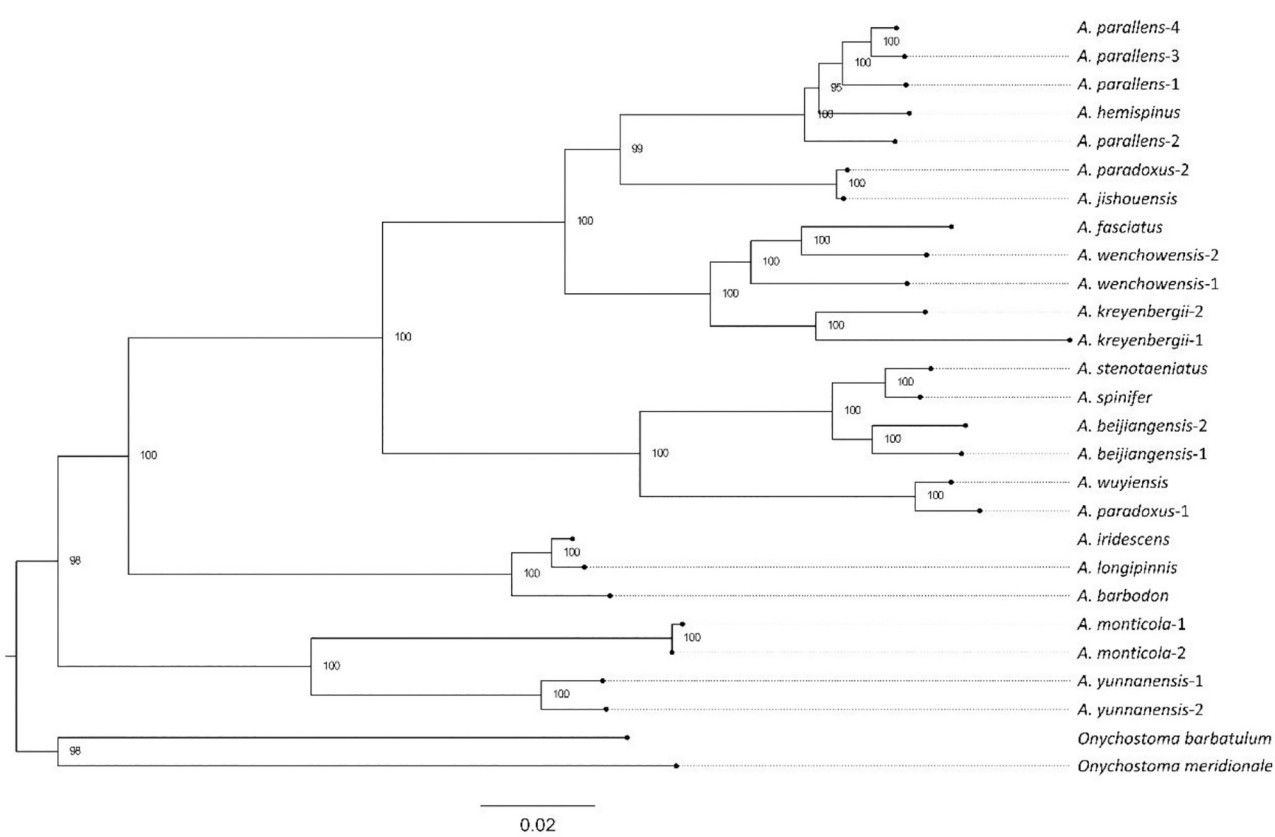

**Fig 3. Maximum-likelihood tree of the *Acrossocheilus* genus based on 13 protein coding mitochondrial gene sequences.** *Onychostoma barbatulum* and *O. barbatulum* were used as outgroup. The different individuals are labeled at the tips. Numbers at the nodes denote the bootstrap values.

## Discussion

### Phylogenetic relationships

The phylogenetic positions resolved for the majority of taxa included in this study were consistent with prior inferences based on traditional taxonomy [12] and molecular markers [38–40]. However, as more accessions included, our analyses revealed some relationships that suggest that species boundaries in *Acrossocheilus* remain unsettled. For example, *A. barbodon*, which is endemic to Hainan Island of southern China, has widely been regarded as a synonym of *A. iridescens*, which occurs in the tropics of Southern China and Southeast Asia. However, in our analyses, *A. iridescens* formed a clade with *A. longipinnis* that was sister to *A. barbodon*. This

**Table 3. Codons that candidates for being under positive selection, based on four selection tests.**

| Gene | MEME | | FUBAR | | FEL | SLAC |
|------|------|---------|-------|-------|------|------|
| | Codon | P-value | Codon | prob | Codon | Codon |
| COX1 | 133 | 0.04 | | | | |
| COX2 | 186 | 0.04 | | | | |
| ND3 | 1 | 0.01 | 1 | > 0.9 | 1 | |
| ND4 | 351 | 0.04 | 51 | > 0.9 | 51 | 51 |
| | 374 | 0.04 | | | | |
| COX3 | 213 | 0.03 | | | | |

suggests that *A. barbodon* may merit species status. Moreover, accessions representing a synonym of *A. longipinnis*, *A. stenotaeniatus*, were unexpectedly recovered in a clade with *A. spinifer*. Thus, the status of *A. stenotaeniatus* with respect to *A. spinifer* is unsettled and in-depth taxonomic work is merited to resolve it. Additionally, the species, *A. hemispinus*, was nested within the clade of *A. parallens*, meaning that the former may be a synonym of the latter. Notably, one individual of *A. paradoxus* formed a clade with *A. jishouensis*, while another individual of this species formed a clade with *A. wuyiensis*, suggesting that *A. paradoxus* may comprise a species complex with cryptic biodiversity.

Two possibilities may explain why our phylogenetic results show several cases of paraphyly and polyphyly of well-accepted species and their synonyms. One possibility is that using mitochondrial genes alone does not reflect the real phylogenetic positions of *Acrossocheilus* species. Alternatively, the taxonomy of this genus may be inadvertently based on convergently evolved or labile morphological characters and is, therefore, inconsistent with molecular phylogenetic relationships. Overall, further taxonomic work on *Acrossocheilus* is merited to resolve species boundaries in the genus as well as to elucidate trait evolution.

## Gene evolution and natural selection

Here, we present a comprehensive comparative analysis of mitogenome evolution in *Acrossocheilus*. The synonymous substitutions and nonsynonymous substitutions in the sampled mitochondrial coding genes evolved in a near neutral manner as is predicted for mitogenomes [41]. In our study, neutral evolution is evidenced by the fact that the numbers of synonymous and nonsynonymous mutations are highly correlated with the length of the respective genes (Fig 2B). However, where selection can be inferred, the Ka/Ks values revealed that purifying selection generally dominates mitochondrial genome evolution. This result was consistent with the corresponding values (Ka/Ks < 1) detected in the mitogenomes of other fishes [3, 9].

The differing mutation rates ($\mu_{Relative}$) that we detected across different genes may result from differences in the strength of purifying selection due to functional constraints [9, 24, 42]. In a prior study, the rate of mutation in mitochondrial genes was shown to be linked to gene position [9]. Specifically, genes such as *NADH* and *CYTB* may evolve more rapidly because they are further from the origin of mitogenomic replication and accumulate more mutations by spending a longer time in the single stranded during replication. This is consistent with our results in which we found that *NADH* and *CYTB* genes generally had a greater number of substitutions that *COX* genes, which are closer to the origin. A higher mutation rate may necessitate stronger purifying selection.

Despite the prevalence of purifying selection in mitogenomes, the possibility for positive selection acting on single codon positions cannot be excluded, and, potentially, can facilitate physiological adaptations to new environments [24, 43]. In this study, the codon-based analyses of selection showed that several sites may be under positive selection. Among the analyses, TreeSAAP revealed the largest number of positive sites, but this program tends to have a high false positive rate [4, 33]. The more conservative approaches, using MEME, SLAC, FUBAR, and FEL revealed far fewer positive sites. Among these, MEME identified six possible sites in five genes compared to even fewer among the other methods. MEME is sensitive to episodic positive selection, while SLAC, FUBAR, and FEL detect pervasive positive selection. Thus, positive selection may be more likely to be episodic in the mitogenomes of *Acrossocheilus* and pervasive positive selection less common. These episodic events could be related to ecological shifts into new environments. However, the lack of strong evidence for positive selection using the CODEML method may suggest a relatively limited role of mitochondria in facilitating adaptations to new environments or show that only limited mitochondrial change is necessary.

Nevertheless, further studies on the genes inferred to have signatures of positive selection are needed at the intraspecific-level to more fully understand their roles in environmental adaptation of this genus.

Overall, although some amino acid positions are subject to positive selection, most of the positions are evolving neutrally or are under purifying selection. The patterns in *Acrossocheilus* appear to be best explained by relaxed purifying selection, and this was supported by the Ka/Ks value for each gene and the large number of sites under purifying selection detected in codon-based methods.

## Relationships of Ka/Ks to the environments of sampled individuals

A few studies have assessed if selection on mitochondrial genes is correlated with environmental variables. For example, environmental analysis for penguins revealed a high correlation between *ND4* and sea surface temperature [44]. Notably, our analysis of the relationship between environmental variables and Ka/Ks ratios revealed that *ATP6* was correlated with mean diurnal temperature range (Bio2), while *COX2* was associated with several precipitation variables. A positive correlation between an environmental variable and Ka/Ks ratio for a specific mtDNA gene may suggest that species inhabiting similar environments undergo convergent adaptative processes for that gene, and that the processes are divergent in different environments.

Environments with high energy demands, such as extreme climates, are typically associated with stronger functional constraints on metabolism; that is, lower Ka/Ks values associated with codons/genes linked to energy metabolism [45]. This is consistent with our findings that the Ka/Ks values for *ATP6* and *COX* genes are generally low relative to those of *ND* genes as the former two genes were found correlated with specific environmental variables, while the latter were not. However, TreeSAAP revealed some significant amino acid changes affecting the equilibrium constant in both *ATP6* and *COX2* as well as the alpha-helical tendencies for *COX2*, even though we did not detect any sites of positive selection in this gene has been found in other pelagic fish [46]. Thus, mutations observed in *ATP6* of *Acrossocheilus* could be related to relaxed purifying selection. In contrast, codon 186 was detected as a possible site of positive selection in *COX2* in the MEME analysis. The *COX2* gene is a catalytic subunit of complex IV of the OXPHOS and plays a role in increasing the coupling efficiency to produce *ATP* and, consequently, heat. Positively selected sites that appear to interact with other *COX* subunits (complex IV) have been reported from other fish, such as *Scombroidei* [11, 46]. Overall, the correlation between precipitation-related variables and Ka/Ks value for *COX2* could be because frequent rains reoxygenate waters [47], thereby increasing the dissolved oxygen available for ATP production.

Although we detected several significant correlations between Ka/Ks ratios and environmental variables, we cannot reject the possibility that these results are influenced by other variables not evaluated here, such as salinity.

## Conclusion

Here, we identify mtDNA candidate genes under selection which could be involved in broad-scale adaptations of *Acrossocheilus* species to their environment.

This is a novel study to comprehend adaptation to the environment occurring at a molecular level. Integration of environmental and molecular data provide insights into how *Acrossocheilus* species have adapted to their environments and therefore how they may respond to future, human-induced changes to their environment.

## Supporting information

**S1 Table. Details of site models used in CodeML.**
(PDF)

**S2 Table. Standard bioclimatic variables for the *Acrossocheilus* individuals in fourteen localities.**
(PDF)

**S3 Table. Significant changes in amino acid properties detected in TreeSAAP.**
(PDF)

**S1 Dataset. Dataset for phylogenetic analysis and analyses of evolutionary rates.**
(DOCX)

## Acknowledgments

The authors would like to thank TopEdit (www.topeditsci.com) for linguistic assistance during preparation of this manuscript.

## Author Contributions

**Data curation:** Yudong Guo.

**Project administration:** Yang Gao.

**Writing – original draft:** Dan Zhao.

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
