## [Decision Letter · Decision Letter 0]

15 Aug 2022

PONE-D-22-15319Natural selection drives the evolution of mitogenomes in AcrossocheilusPLOS ONE

Dear Dr. Gao,

Thank you for submitting your manuscript to PLOS ONE. After careful consideration, we feel that it has merit but does not fully meet PLOS ONE’s publication criteria as it currently stands. Therefore, we invite you to submit a revised version of the manuscript that addresses the points raised during the review process.

We look forward to receiving your revised manuscript.

Kind regards,

Bi-Song Yue, Ph.D

Academic Editor

PLOS ONE

Reviewers' comments:

Reviewer's Responses to Questions

**Comments to the Author**

1. Is the manuscript technically sound, and do the data support the conclusions?

Reviewer #1: Partly

Reviewer #2: Partly

2. Has the statistical analysis been performed appropriately and rigorously? 

Reviewer #1: N/A

Reviewer #2: Yes

3. Have the authors made all data underlying the findings in their manuscript fully available?

Reviewer #1: Yes

Reviewer #2: Yes

4. Is the manuscript presented in an intelligible fashion and written in standard English?

Reviewer #1: No

Reviewer #2: No

5. Review Comments to the Author

Reviewer #1: 1. The current work cannot fully support "ATP6 was significantly correlated with a mean diurnal temperature range" and "Ka/Ks ratio of COX2 was strongly associated with several precipitation-related variables". The authors are required to provide further analysis or sufficient evidence to support these views.

2. The authors declared that their work provides a foundation for integrating the mitochondrial genome into future investigations of how these species may respond to ongoing global climatic changes. I don’t understand how did they conclude that?

3. The author selected 25 species with the closest relationship to construct a phylogenetic tree, just for selection analysis? When the author performed phylogenetic analysis and Ka/Ks analysis, it is necessary to expand the research scope of species, and compare the differences between different taxa in combination with the traits of species. Obviously, this research is not enough.

4. Lines (78-82), please provide a bibliography or citation source.

5. Many researches have focused on the environmental adaptation of organisms. A previously published mitochondrial study (DOI: https://doi.org/10.1186/s12862-021-01803-y) can provide a reference for this study.

6. It is recommended that the results of the selection analysis should be presented graphically.

7. Line 307, "<<" should be "<".

8. Lines (309-311) “The differing mutation rates (μRelative) that we detected across different genes may result from differences in the strength of purifying selection due to functional constraints”. Is there sufficient evidence to support this conclusion? If yes, please list it. If not, please delete it.

Reviewer #2: This is fine study, and some data is valuable. However, I am not quite sure that the current study fits to the journal scope and standard because this is very general study which is not something new.

1. “This genus has 26 species, which have a center of diversity (21 species) in southern China and commonly occur in middle and/or lower reaches of river drainage”. But in this paper, the authors only chose 25 published sequences of 14 species to analyze. It is better to add more new sequenced species to confirming the findings.

2. In this paper, although the phylogenetic tree with high bootstrap, if we add other species, the topology will change, and the authors did not describe the taxonomy of the genus in the introduction.

6. PLOS authors have the option to publish the peer review history of their article (what does this mean?). If published, this will include your full peer review and any attached files.

Reviewer #1: No

Reviewer #2: No

---

## [Author Response · Author response to Decision Letter 0]

8 Sep 2022

Reviewer #1: 1. The current work cannot fully support "ATP6 was significantly correlated with a mean diurnal temperature range" and "Ka/Ks ratio of COX2 was strongly associated with several precipitation-related variables". The authors are required to provide further analysis or sufficient evidence to support these views.

Answer: We have revised the manuscript and adopted a euphemism.

2. The authors declared that their work provides a foundation for integrating the mitochondrial genome into future investigations of how these species may respond to ongoing global climatic changes. I don’t understand how did they conclude that?

Answer: We have revised this sentence to “Based on this, we believe that our study provides a new insight into the role of the mitochondrial genome of Acrossocheilus species in adaptation to different environments.”

3. The author selected 25 species with the closest relationship to construct a phylogenetic tree, just for selection analysis? When the author performed phylogenetic analysis and Ka/Ks analysis, it is necessary to expand the research scope of species, and compare the differences between different taxa in combination with the traits of species. Obviously, this research is not enough.

Answer: We have revised the manuscript. We included 25 published mitochondrial genome sequences from 14 species in NCBI. The remaining species, although reported, are difficult to sample and accurately identify due to their rarity and existence of synonyms with species that included in this study (A. rendahli may as a synonym for A. yunnanensis) [12]. Besides, there are some species that may in fact belong to other genera, such as, A.malacopterus may belong to the genus Onychostoma and A. ikedai should probably belong to the genus Poropuntius [12]. Therefore, only 25 published mitochondrial genome sequences from 14 species in this genus were included in this study. Moreover, in another study, we are trying to address the taxonomic confusion of species not included in this study using the method of molecular delimitation of species.

[12]. Yuan LY. Taxonomic Revision of Chinese Species of the Cyprinid Genus Acrossocheilus (Teleostei: Cypriniformes). M.Sc. Thesis, Nanchang University. 2005. Available from: https://kns.cnki.net/kcms/detail/detail.aspx?dbcode=CMFD&dbname=CMFD0506&filename=2006022941.nh&uniplatform=NZKPT&v=SzhvHT2d3k6d13FM3lvMO5bpIg77SwIPRiq0vxjXRGcmjD6JnQ8_-LnSzQBqH_Ci

4. Lines (78-82), please provide a bibliography or citation source.

Answer: We have added citations and references to the manuscript (doi: 10.1186/1471-2164-9-119; doi: 10.1111/mec.12240; doi: 10.1111/jzs.12079).

5. Many researches have focused on the environmental adaptation of organisms. A previously published mitochondrial study (DOI: https://doi.org/10.1186/s12862-021-01803-y) can provide a reference for this study.

Answer: Thank you for your reference.

6. It is recommended that the results of the selection analysis should be presented graphically.

Answer: We have displayed the results of the selection analysis in a graph, named Fig1.

7. Line 307, "<<" should be "<".

Answer: We have revised the manuscript.

8. Lines (309-311) “The differing mutation rates (μRelative) that we detected across different genes may result from differences in the strength of purifying selection due to functional constraints”. Is there sufficient evidence to support this conclusion? If yes, please list it. If not, please delete it.

Answer: We have added citations and references to the manuscript (doi: 10.1016/j.ympev.2015.11.008; doi: 10.1016/j.tig.2007.03.008; doi: 10.1093/oxfordjournals.molbev.a004014).

Reviewer #2: This is fine study, and some data is valuable. However, I am not quite sure that the current study fits to the journal scope and standard because this is very general study which is not something new.

1. “This genus has 26 species, which have a center of diversity (21 species) in southern China and commonly occur in middle and/or lower reaches of river drainage”. But in this paper, the authors only chose 25 published sequences of 14 species to analyze. It is better to add more new sequenced species to confirming the findings.

Answer: We have revised the manuscript. We included 25 published mitochondrial genome sequences from 14 species in NCBI. The remaining species, although reported, are difficult to sample and accurately identify due to their rarity and existence of synonyms with species that included in this study (A. rendahli may as a synonym for A. yunnanensis) [12]. Besides, there are some species that may in fact belong to other genera, such as, A.malacopterus may belong to the genus Onychostoma and A. ikedai should probably belong to the genus Poropuntius [12]. Therefore, only 25 published mitochondrial genome sequences from 14 species in this genus were included in this study. Moreover, in another study, we are trying to address the taxonomic confusion of species not included in this study using the method of molecular delimitation of species.

[12]. Yuan LY. Taxonomic Revision of Chinese Species of the Cyprinid Genus Acrossocheilus (Teleostei: Cypriniformes). M.Sc. Thesis, Nanchang University. 2005. Available from: https://kns.cnki.net/kcms/detail/detail.aspx?dbcode=CMFD&dbname=CMFD0506&filename=2006022941.nh&uniplatform=NZKPT&v=SzhvHT2d3k6d13FM3lvMO5bpIg77SwIPRiq0vxjXRGcmjD6JnQ8_-LnSzQBqH_Ci.

2. In this paper, although the phylogenetic tree with high bootstrap, if we add other species, the topology will change, and the authors did not describe the taxonomy of the genus in the introduction.

Answer: Our phylogenetic analyses produced identical topologies with previous studies [38]. Moreover, we have added the distribution of this genus in different temperature zones in the introduction. 

[38]. Yuan LY, Liu XX, Zhang E. Mitochondrial Phylogeny of Chinese Barred Species of the Cyprinid Genus Acrossocheilus Oshima, 1919 (Teleostei: Cypriniformes) and Its Taxonomic Implications. Zootaxa. 2015; 4059: 151–168. https://doi.org/10.11646/zootaxa.4059.1.8.

---

## [Editor Report · Decision Letter 1]

29 Sep 2022

Natural selection drives the evolution of mitogenomes in Acrossocheilus

PONE-D-22-15319R1

Dear Dr. Gao,

We’re pleased to inform you that your manuscript has been judged scientifically suitable for publication and will be formally accepted for publication once it meets all outstanding technical requirements.

Kind regards,

Bi-Song Yue, Ph.D

Academic Editor

PLOS ONE

---

## [Editor Report · Acceptance letter]

4 Oct 2022

PONE-D-22-15319R1 

Natural selection drives the evolution of mitogenomes in *Acrossocheilus*

Dear Dr. Gao:

I'm pleased to inform you that your manuscript has been deemed suitable for publication in PLOS ONE. Congratulations! Your manuscript is now with our production department. 

Kind regards, 

on behalf of

Dr. Bi-Song Yue 

Academic Editor

PLOS ONE